# Chitosan-Functionalized Poly(β-Amino Ester) Hybrid System for Gene Delivery in Vaginal Mucosal Epithelial Cells

**DOI:** 10.3390/pharmaceutics16010154

**Published:** 2024-01-22

**Authors:** Xueqin Gao, Dirong Dong, Chong Zhang, Yuxing Deng, Jiahui Ding, Shiqi Niu, Songwei Tan, Lili Sun

**Affiliations:** 1Department of Pharmacy, Union Hospital, Tongji Medical College, Huazhong University of Science and Technology, Wuhan 430022, China; 2018xh0041@hust.edu.cn; 2Department of Obstetrics and Gynecology, Zhongnan Hospital, Wuhan University, Wuhan 430062, China; dongdirong@znhospital.cn; 3Tongji School of Pharmacy, Tongji Medical College, Huazhong University of Science and Technology, Wuhan 430030, China; m201875232@alumni.hust.edu.cn (C.Z.); dengyuxing@simm.ac.cn (Y.D.); m201975433@alumni.hust.edu.cn (J.D.); m201973061@hust.edu.cn (S.N.); 4Health Bureau of Luannan Country, Tangshan 063599, China

**Keywords:** gene delivery, poly(β-amino ester), chitosan, mucoadhesion

## Abstract

Gene therapy displays great promise in the treatment of cervical cancer. The occurrence of cervical cancer is highly related to persistent human papilloma virus (HPV) infection. The HPV oncogene can be cleaved via gene editing technology to eliminate carcinogenic elements. However, the successful application of the gene therapy method depends on effective gene delivery into the vagina. To improve mucosal penetration and adhesion ability, quaternized chitosan was introduced into the poly(β-amino ester) (PBAE) gene-delivery system in the form of quaternized chitosan-g-PBAE (QCP). At a mass ratio of PBAE:QCP of 2:1, the polymers exhibited the highest green fluorescent protein (GFP) transfection efficiency in HEK293T and ME180 cells, which was 1.1 and 5.4 times higher than that of PEI 25 kD. At this mass ratio, PBAE–QCP effectively compressed the GFP into spherical polyplex nanoparticles (PQ–GFP NPs) with a diameter of 255.5 nm. In vivo results indicated that owing to the mucopenetration and adhesion capability of quaternized CS, the GFP transfection efficiency of the PBAE–QCP hybrid system was considerably higher than those of PBAE and PEI 25 kD in the vaginal epithelial cells of Sprague–Dawley rats. Furthermore, the new system demonstrated low toxicity and good safety, laying an effective foundation for its further application in gene therapy.

## 1. Introduction

Cervical cancer is one of the most common gynecologic tumors and a serious threat to the health of women. Persistent human papilloma virus (HPV) infection due to the integration of the viral genome into the host genome is an essential factor underlying the occurrence and development of cervical cancer [1]. Achieving complete viral eradication is the key to treating HPV infection. The fast-paced development of genome editing technologies, especially the CRISPR/Cas9 system, is a potential treatment method for viral infection-associated cancer, such as cervical cancer [2].

Successful gene therapy depends on the safe and efficient delivery of negatively charged and easily degradable nucleic acids to targeted cancer cells by a gene vector. Although viral vectors exhibit higher transfection efficiency (TE), they are associated with a few inherent drawbacks, such as increased immunogenicity and potential genotoxic effects [3,4]. Furthermore, in a few cases, viral vectors may even activate a proto-oncogene or inactivate a tumor suppressor gene, leading to tumor development [5,6]. Compared with viral vectors, nonviral vectors demonstrate lower immunogenicity and provide higher safety, and, hence, these vectors have received more attention in different gene-delivery systems [7]. During the implementation of gene therapy for cervical cancer, vaginal administration is advantageous owing to higher therapeutic efficacy and fewer side effects compared with systemic administration [8,9]. However, vaginal mucus acts as a major barrier as it reduces the adhesion of the gene therapy system and obstructs the diffusion of the plasmid required for gene delivery [10,11]. Therefore, a delivery system that can overcome these effects of mucus, prolong the drug retention time, and enhance therapeutic efficacy is desirable [12].

Chitosan (CS) is a natural cationic polymer with mucosal adhesion capability, low toxicity, and biocompatibility. CS has been applied in numerous mucosal drug delivery systems [13,14]. Hydrogen bonding, hydrophobic interactions, and electrostatic interactions between CS and mucous layer/mucosal epithelial cells can extend the retention time of the system, maintain local drug concentration, and achieve better bioavailability [15,16,17]. However, the TE of CS is affected by its molecular weight and deacetylation degree, and these factors limit the application of CS in gene-delivery systems [18,19,20]. Notably, CS can be further modified via methods such as quaternization and copolymerization [21]. Quaternized chitosan is a water-soluble derivative of chitosan that exhibits improved mucoadhesivity and permeability-enhancing properties [22,23]. Some studies have shown that quaternized chitosan can open the tight junctions of intestinal epithelial cells to allow the paracellular transport of hydrophilic molecules and can be used as an intestinal epithelial absorption enhancer [24]. However, quaternization may lead to difficulty in DNA dissociation due to its excessive DNA condensation ability, thereby limiting gene expression [25].

Poly(β-amino ester) (PBAE)-based nonviral vector is associated with several characteristics, such as biodegradability, low production cost, and easily modifiable chemical structure [26,27]. Many high-throughput in vitro screening studies of gene vector libraries based on PBAE have shown that PBAE can achieve high-level transgene expression, including many challenging biological barriers, such as the entire luminal surface of the mouse lungs [28,29,30]. Additionally, PBAE with a disulfide bond skeleton can promote degradation, release DNA within cells, and reduce cytotoxicity. Previously, we synthesized a series of PBAE-based copolymers and applied them in the delivery of chemotherapy drugs [31] and nucleic acid. More importantly, we achieved the delivery of CRISPR/Cas9 to HPV-infected cancer cells [32].

To improve mucosal adhesion and gene transfection ability, we prepared a cationic copolymer quaternized CS-g-poly(β-amino ester) (QCP) and determined the optimal mass ratio (weight ratio) of linear PBAE and QCP by evaluating its TE in normal (HEK293T) and cervical cancer (ME180) cell lines. We made further modifications by compounding the copolymer with green fluorescent protein (GFP) plasmid (at the best mass ratio) to form the optimal polymer nanoparticles (NPs) PQ–GFP NPs. The particle size, ζ-potential, stability, morphology, DNA-condensing ability, cytotoxicity, vaginal mucoadhesion rate of the hybrid system, and TE in other cell lines were measured. Moreover, the delivery effectiveness and safety of the system in vivo were verified through in situ local administration.

## 2. Materials and Methods

### 2.1. Materials

1,4-butanediol diacrylate (BDD), 5-amino-1-pentanol (C32), 1,3-propanediamine, and dehydrated glyceryl trimethylammonium chloride (GTMAC) were purchased from TCI (Shanghai, China). CS was obtained from Golden-Shell Pharmaceutical Co., Ltd. (Yuhuan, China) with a viscosity-average molecular weight of ~20,000, with a 95% degree of deacetylation. Dithiothreitol (DTT),1-ethyl-(3-dimethylaminopropyl) carbodiimide (EDC), N-hydroxysuccinimide (NHS), N, N′-dicyclohexylcarbodiimide (DCC), and maleimide butyric acid (MAL) were purchased from Aladdin Bio-Chem Technology Co., Ltd. (Shanghai, China). All the solvents used were of analytical grade and obtained from Sinopharm Chemical Reagent Co., Ltd. (Shanghai, China). Dulbecco’s Modified Eagle’s medium (DMEM), penicillin–streptomycin, fetal bovine serum (FBS), and trypsin were purchased from Gibco^®®^ (Thermo Fisher Scientific^TM^, Waltham, MA, USA).

The cells were cultured in complete DMEM supplemented with 10% FBS at 37 °C in the presence of 5% CO_2_. At 80% cell fusion, trypsin was added and DMEM was changed every other day.

Female Sprague–Dawley (SD) rats (weighing ~200 g) were purchased from Beijing Vital River Laboratory Animal Technology Co., Ltd. (Beijing, China). All experimental animals were raised under the guidance of the Huazhong University of Science and Technology Laboratory Animal Care and Use Guidelines and approved by the Experimental Animal Ethics Committee of Tongji Medical College of Huazhong University of Science and Technology.

### 2.2. QCP Synthesis

The syntheses of quaternized chitosan (QCS), quaternized chitosan maleimide butanamide (Mal-QCS), and thiol terminal PBAE (SH-PBAE) are described in the Appendix A. QCP was synthesized via the click chemistry of Mal–QCS and SH–PBAE (Figure 1A). Briefly, QCS was synthesized following the procedure described in our previous work [33]. Mal–QCS was synthesized using the amidation reaction of QCS with maleimide butyric acid. Acrylate-terminated PBAE was synthesized following the procedure described in our previous study [32] and reacted with excess. Subsequently, Mal–QCS and SH–PBAE were successfully conjugated in acetate buffer (pH 5.0) for 4 h in an ice bath. The reaction solution was subsequently precipitated in acetone to remove excessive SH–PBAE and dried under vacuum to obtain the QCP. The chemical structures of QCP and other intermediate products were characterized using proton nuclear magnetic resonance (^1^H-NMR; Bruker AVANCE III 400 MHz NMR spectrometer; solvents, D_2_O-d-trifluoroacetic acid).

To verify the feasibility of thiol-Michael addition reaction in acetate buffer (pH = 5.0), we conducted a reaction between MAL and DTT (Appendix A). The reaction details are described in the Appendix A. The structure and molecular weight of the final product were verified via ^1^H-NMR and mass spectrometry (MS; Bruker Daltonics microOTOF II spectrometer, Billerica, MA, USA).

### 2.3. Preparation and Characterization of PQ–GFP Polyplex NPs

The model plasmid containing the GFP gene was used to evaluate the TE of different cationic polymers. To prepare PBAE–QCP–GFP polyplex NPs (PQ–GFP NPs), PBAE and QCP in different proportions (0:1, 0.5:1, 1:1, 2:1, 4:1, and 1:0) were dissolved together in acetate buffer (pH 5.0) and then mixed with GFP at predetermined mass ratios (30:1, 50:1, 75:1, and 100:1), followed by vortexing for ~30 s and incubation for 30 min. PEI–GFP polyplex NPs were prepared following the same method. The nitrogen/phosphorus (N/P) ratios of different polymers to GFP are presented in Appendix A. The particle size and ζ-potential of synthesized NPs were determined using dynamic light scattering (DLS, Brookhaven, Suffolk, NY, USA). The stability of NPs in FBS was measured through monitoring the particle size changes at different time intervals (0.5 h, 1 h, 2 h, 4 h, 8 h, 12 h, 24 h, and 48 h). The morphology of NPs was studied using transmission electron microscopy (TEM, HT7800, Tokyo, Japan). Agarose gel electrophoresis was performed to determine the effective compression of plasmids.

### 2.4. Optimization and Characterization of PQ–GFP NPs

The optimal mass ratio of PBAE and QCP was first investigated in human embryonic kidney (HEK293T) and cervical cancer (ME180) cell lines. Briefly, HEK293T/ME180 cells were transferred to six-well plates (2 × 10^5^ cells per well) and cultured at 37 °C in the presence of 5% CO_2_ for 24 h. Subsequently, DMEM was removed and the cells were washed with phosphate-buffered saline (PBS). Finally, each well was supplied with 2 mL DMEM without serum. Sterile PBAE–QCP solutions with different PBAE–QCP mass ratios were mixed with GFP at different mass ratios, incubated at room temperature for 30 min, and added into six-well plates. After 4 h, the medium was replaced with DMEM containing 10% FBS. After 36 h of incubation, the TE of each group was qualitatively analyzed via fluorescence microscopy (CKX53, Olympus, Tokyo, Japan). Subsequently, these cells were washed with PBS, trypsinized, dispersed in PBS, and monitored via flow cytometry (Accuri^®^C6, BD, Franklin, NJ, USA) to quantitatively analyze the TE. The TE of PQ–GFP NPs in other cervical cancer cell lines (SiHa and HeLa), as well as that of PEI–GFP NPs in the corresponding cell lines, was evaluated using the same method.

### 2.5. MTT Analysis

To evaluate the safety of PQ–GFP NPs for ME180 cells, an MTT assay was performed. Briefly, the cells were seeded in 96-well plates (8 × 10^3^ cells per well), and after overnight attachment, the medium was discarded. Various concentrations of PQ or PQ–GFP NPs were subsequently added. Following 36 h of incubation, 10 μL of MTT solution (5 mg∙mL^−1^) was added into each well, followed by culturing for 2 h. Subsequently, the medium was discarded and 150 μL of dimethyl sulfoxide was added to dissolve the precipitate. A microplate reader (Thermo, Waltham, MA, USA) was employed to record the absorbance of each well at 490 nm. Five replicate wells were used for each sample to obtain the average value and standard deviation.

### 2.6. In Vitro Mucosal Adhesion of PQ–GFP NPs

In vitro mucosal adhesion of PQ–GFP NPs was evaluated in the vaginal tissue of rats. PBAE was labeled using Ce6, a fluorescent molecule, as described previously [32]. The vaginal tissues (2.28 g) of the rats washed with normal saline were immersed in PBAE–Ce6–GFP (1 mg∙mL^−1^), PBAE–Ce6/QCP–GFP NPs (1 mg∙mL^−1^), or free Ce6, with an equivalent amount of Ce6. After 3 h of incubation at 37 °C, the tissues were washed using saline to remove unattached materials and then soaked in anhydrous ethanol for another 10 min at 37 °C to re-dissolve the Ce6 adhered to the tissues. The fluorescence intensities of Ce6 before and after adhesion were measured via fluorescence spectrophotometry (F-4600, Tokyo, Japan; excitation: 633 nm; emission: 660 nm) and the relative vaginal mucoadhesion rate was calculated as follows (set Ce6 group as 1).

### 2.7. Transfection Ability in the Vagina of SD Rats

Female SD rats weighing ~200 g were used to investigate the in vivo transfection ability following the vaginal administration of normal saline, PEI–GFP, PBAE–GFP, QCP–GFP, or PQ–GFP NPs (n = 3 per group). The mass ratios of PBAE and QCP in PQ–GFP NPs and PBAE–QCP to GFP were 2:1 and 75:1, respectively. The GFP dosage was 40 μg per rat. The rats were administered with different compounds once a day for 3 consecutive days, following which their vaginal epithelial cells were collected on the 6th day. The collected cells were stained with DAPI and observed under a fluorescence microscope. Furthermore, the TE was estimated using flow cytometry. In addition, the frozen section images of SD rat vaginas were captured via fluorescence microscopy.

### 2.8. Toxicity Analysis of PQ–GFP NPs in SD Rats

The systemic toxicities of PEI–GFP, PBAE–GFP, and PQ–GFP NPs in SD rats were assessed using the in situ vaginal administration method. The dosage and method of administration were same as described in Section 2.7. On the 6th day, blood and serum were collected for routine blood and biochemical tests and these included white blood cells (WBCs), red blood cells (RBCs), hemoglobin (HGB), mean corpuscular volume (MCV), mean corpuscular hemoglobin (MCH), platelets (PLTs), alanine aminotransferase (ALT), aspartate transaminase (AST), alkaline phosphatase (ALP), blood urea nitrogen (BUN), glucose (GLU), and total cholesterol (CHO). Additionally, the vagina along with other organs (cervix, uterus, ovary, urethra, rectum, colon, heart, liver, spleen, lung, and kidney) of rats were harvested, fixed with 4% paraformaldehyde, and sliced for hematoxylin–eosin (H&E) staining.

## 3. Results

### 3.1. Synthesis and Characterization of QCP

Figure 1B reveals the ^1^H-NMR results of QCP. The chemical shifts at 2.0, 3.2, and 4.5 ppm indicate the hydrogen signal peaks of GTMAC in QCS. In addition, a hydrogen signal peak on the CS occurred at 3.5–4.0 ppm, indicating the successful synthesis of QCS. The signal at 6.8 ppm belonged to the hydrogen on the double bond in the five-membered ring of maleimidobutyric acid in Mal–QCS. The chemical shift value at 4.06 ppm belonged to the hydrogen signal peak of –COOCH_2_– on the BDD unit of PBAE, and the signal at 1.53 to 1.32 ppm was from –CH_2_– on the C32 moiety. In addition, the chemical shift at 5.3 ppm was from the hydroxyl group on DTT in SH–PBAE. The disappearance of the signal at 6.8 ppm (in Mal–QCS) in QCP, which belonged to alkene bonds on the acrylic esters’ end group, confirms the successful conjugation between acrylic esters and the SH group. To further assess the click chemistry, we mixed MAL and DTT in acetate buffer (pH = 5.0). As presented in MS (Appendix A), two peaks with molecular weights of 360 and 543 were clearly detected. ^1^H-NMR (Appendix A) also indicated that the double bond in the five-membered ring of MAL decreased to 67% in the final product. These results proved that the thiol–alkene click chemistry between DTT and MAL can perform under weak acid conditions. Finally, by comprehensively comparing the spectra of QCP and each intermediate product, it could be determined that QCP had been successfully synthesized.

### 3.2. Preparation and Optimization of PQ–GFP NPs

As a hybrid system, the mass ratio of PBAE and QCP was an important parameter for TE. Conversely, cationic carriers may be cytotoxic to cells, owing to their high charge density [34]. Therefore, identifying the optimal mass ratio of PBAE and QCP and an appropriate proportion of cationic polymers and DNA is necessary to achieve high gene TE and low cytotoxicity. PEI 25 kD, considered the “gold standard” in gene vector, was selected as a positive control. GFP was employed as a model plasmid to evaluate the performance of the gene-delivery vectors [35]. The vector map of pGFP is presented in Appendix A.

First, the optimal mass ratio of PEI and GFP plasmid was explored in HEK293T and ME180 cells via transfection experiments (Appendix A). The optimal PEI:GFP mass ratios were 7.5:1 and 45:1 in HEK293T and ME180 cells. The CS and QCS alone could hardly deliver GFP in the ME180 cell line (Appendix A). Based on our previous studies, the PQ–GFP nanoparticles were prepared with different mass ratios of PBAE and QCP, and the mass ratio of PBAE–QCP and GFP was selected as 75:1. Following transfection in HEK293T cells for 36 h, GFP expression varied with the mass ratio of PBAE and QCP (Figure 2A,B, Appendix A). QCP only demonstrated a TE of 3.0%, probably because of the low transfection ability of QCS (Appendix A). Conversely, PBAE displayed a TE of 97.4%. For the PBAE–QCP hybrid system, the TEs were 91.3%, 97.0%, 97.4%, and 97.9% when the mass ratios of PBAE and QCP were 0.5:1, 1:1, 2:1, and 4:1, respectively, higher than those of the PEI with the best material/plasmid mass ratio (Appendix A). Living cell count (LCC) is a parameter that is related to the cytotoxicity of the gene transfection system. The relative LCC was >0.6, indicating low cytotoxicity of all the cationic polymers (Figure 2C). Compared with the 4:1 mass ratio, which exhibited the highest TE among the four PBAE–QCP hybrid polymers in HEK293T cells, the 2:1 system demonstrated nearly equal TE but substantially higher LCC. We further tested TE and LCC of the polymers in the ME180 cell line (Figure 2D–F, Appendix A). Surprisingly, the 2:1 system displayed the highest TE and LCC in ME180 cells. Taking TE and LCC into consideration, the PBAE–QCP mass ratio of 2:1 was selected for the subsequent studies.

The mass ratio of cationic polymers to GFP also affects the TE. Therefore, the optimum polymer/GFP mass ratio was investigated in HEK293T cells (Figure 3A) and ME180 cells (Figure 3B). When the polymer/GFP mass ratio was 75:1, the optimal TE (98.0% in HEK293T cells and 37.4% in ME180 cells) and the highest relative LCC (0.85 in HEK293T cells and 0.48 in ME180 cells) were observed, consistent with the results of our previous study [32]. A polymer/GFP mass ratio of 75:1 was established in the PQ–GFP NPs.

### 3.3. Characterization of PQ–GFP NPs

PQ–GFP NPs were prepared to evaluate particle size, ζ-potential, stability, morphology, plasmid compression ability, cytotoxicity, and mucosal adhesion. To further explore the correlation between ζ-potential and TE, we examined the ζ-potential of PQ-GFP NPs, as detailed in Appendix A. Notably, QCP exhibited the highest potential due to the presence of ample positively charged quaternary amino groups. As the proportion of PBAE increased, the potential displayed a descending trend overall. Nevertheless, upon combining PQ with GFP at various mass ratios, the potential progressively augmented along with the elevation of the PQ ratio. The optimal PQ–GFP NPs were measured via DLS for particle size and ζ-potential, with a hydrated diameter of 255.5 ± 8.6 nm and a potential of 28.9 ± 4.2 mV (Figure 4A). The size decreased in FBS, possibly because of the electrostatic interaction between NPs and serum protein; however, the stability of NPs remained good for 48 h (Figure 4B). A well-distributed dark circular sphere was observed via TEM, with a particle size of ~170 nm (Figure 4C), which was smaller than the hydrated size measured via DLS. Agarose gel electrophoresis (Figure 4D) of the PQ–GFP NPs with different mass ratios revealed that there was no DNA band when the mass ratio reached 30:1, and free GFP was observed following heparin treatment, indicating that GFP plasmids were successfully compressed by PBAE–QCP via an electrostatic force, thereby avoiding the degradation of GFP plasmid by enzymes and facilitating nanoparticle uptake. As cationic polymers, PBAE and QCP exhibited some cytotoxicity; however, when they were combined with GFP plasmid to form PQ–GFP NPs, the toxicity was obviously reduced (Figure 4E). This may be because of the negative charge of GFP plasmids neutralizing some of the positive charges of the polymer. In addition, the mucoadhesion rates of PBAE–Ce6 and PBAE–Ce6/QCP were 2.7 and 5.2 times higher, respectively, than that of free Ce6 (Figure 4F), confirming that the introduction of QCP increased the local mucosal adhesion of the system in the vagina of SD rats.

We further tested the TE of PQ–GFP NPs in other human cervical cancer lines, SiHa and HeLa (Figure 5). PQ–GFP NPs exhibited good transfection in these cells, which were all superior to PEI–GFP NPs. The TE of PQ–GFP NPs was 3.0 times higher in SiHa cells and 2.1 times higher in HeLa cells compared with that of PEI–GFP NPs. The cells demonstrated good morphology, intact structure, high fusion degree, and no obvious cytotoxicity. Although there are some differences in TE among different cell lines, PQ–GFP NPs exhibited excellent transfection performance and were generally effective for different cell lines, laying the foundation for later application in other fields.

### 3.4. In Vivo Transfection Effect

In treating cervical cancer, local administration is an effective method of increasing the concentration of plasmids in tumor tissues and minimizing undesirable systematic distribution, thereby improving safety [36]. Moreover, the polymer PBAE–QCP could not only prevent the degradation of plasmids by enzymes in vaginal mucus but could also enhance mucosa adhesion. Hence, the PQ–GFP NPs were prepared to transfect the vaginal/cervix epithelial cells of SD rats. The results of fluorescence microscopy (Figure 6A) and flow cytometry (Figure 6B) revealed that the TE of PBAE was 26.7%, which was 1.8 times higher than that of PEI. For the PQ hybrid system, a higher TE was observed, which was 1.4 and 4.8 times higher than those of PBAE and QCP, respectively. Frozen section images (Figure 6C) further revealed that PQ successfully expressed GFP in vaginal or cervical epithelial cells, and the fluorescence expression was the strongest in the PQ group, consistent with the flow data. Notably, the TE of QCP alone was very limited in vitro and in vivo, and the TE of PQ was slightly lower than that of PBAE in the HEK293T cell line. Thus, the enhanced in vivo transfection ability of PBAE–QCP can be attributed to the mucosal adhesion of QCP, which prolonged the retention time of the nanoparticles.

### 3.5. Biosafety of PQ–GFP NPs

In vivo toxicities of PEI–GFP, PBAE–GFP, and PQ–GFP NPs were detected through in situ vaginal administration in SD rats. The values of WBC, RBC, HGB, MCV, MCH, and PLT in routine blood tests were within the normal range, and no statistically significant difference was observed among the groups (*p* > 0.05, Figure 7A). Blood biochemical tests including ALT, AST, ALP, BUN, GLU, and CHO revealed similar results (Figure 7B). In addition, further histopathological examination of the vagina and adjacent organs (such as the cervix, uterus, ovary, urethra, rectum, and colon) as well as other important organs of the whole body (such as the heart, liver, spleen, lung, and kidney) revealed no significant differences in cell morphology, inflammatory cell infiltration, and the presence of obvious blood cells (Figure 7C and Appendix A). All results indicated a good biosafety of the hybrid system for vaginal local administration and its promise for further application in clinical trials.

## 4. Discussion

Chitosan has limited solubility under physiological conditions, low DNA binding efficiency, and poor DNA release properties, making it challenging for gene delivery [37]. However, QCS shows low toxicity to human cells and exhibits strong DNA binding affinity and protection [38]. It outperforms low-molecular-weight chitosan in terms of gene delivery efficiency [39]. PBAE, a highly effective gene delivery vector, has strong DNA binding ability and protects DNA from enzymatic degradation. Its positive charges enable escape from endosomes/lysosomes, promoting DNA entry into the cytoplasm and ensuring effective expression [40].

When QCS was introduced into PBAE, we expected higher transfection efficiency due to the branched segment. However, the TE of QCP was lower than that of free PBAE. We observed, as seen in Appendix A, that CS and QCS alone performed almost no transfection in ME180 cells, likely related to the molecular weight and quaternization degree [24], and the abundant positive charge of individual QCS made it challenging for DNA dissociation and gene expression [25]. Additionally, the effect of QCP was improved but still not as high as free PBAE. According to Appendix A, despite possessing the highest potential, QCP did not achieve the best transfection results, possibly due to spatial hindrance from QCS addition or hydrogen bonding/hydrophobic interaction between QCP and PBAE, which hindered cell uptake. Furthermore, while the high potential strongly bound to DNA, it was not conducive to DNA release within cells. When QCP was further mixed with free PBAE, the proton sponge effect of PBAE enhanced DNA release at specific sites. The strongest effect was achieved at a 2:1 mass ratio in ME180 cells (Figure 2), balancing PBAE’s transfection efficiency and QCS’s mucosal adhesive property, laying the foundation for a local mucosal adhesive system in the SD rat vaginal system.

The in vitro mucosal adhesion experiment confirmed that PQ exhibited a higher vaginal mucoadhesion rate than PBAE (Figure 4F). Moreover, the TE of QCP alone in vivo was limited. While the TE of PQ on cells may be similar to PBAE (Figure 2A,B, Appendix A), it displayed significantly better TE in vivo due to the mucosal adhesion properties of QCP (Figure 6). Local administration not only enhanced drug concentration but also minimized side effects on other parts, enhancing medication safety (Figure 7). Overall, the hybrid system for local vaginal administration demonstrated good biosafety and was well suited for further clinical trials.

## 5. Conclusions

In summary, quaternized CS functionalized PBAE was synthesized and applied as a nonviral gene vector. At a PBAE:QCP mass ratio of 2:1 and material/plasmid mass ratio of 75:1, the hybrid system exhibited the best TE in HEK293T and ME180 cells, with a diameter of 255.5 nm and a potential of 28.9 mV. The hybrid system was stable in FBS and exhibited low cytotoxicity and a superior vaginal mucoadhesion rate. PQ–GFP NPs also demonstrated 3.0 and 2.1 times higher TE than those of PEI in SiHa and HeLa cell lines, respectively. An in situ transfection experiment in the vagina of SD rats confirmed that the introduction of QCP in the PBAE system improved the in situ TE by 1.4 times without cytotoxicity compared with that improved by PBAE alone. Furthermore, good biosafety was observed, laying an optimal foundation for the future clinical transformation of this modified system.

## Figures and Tables

**Figure 1 pharmaceutics-16-00154-f001:**
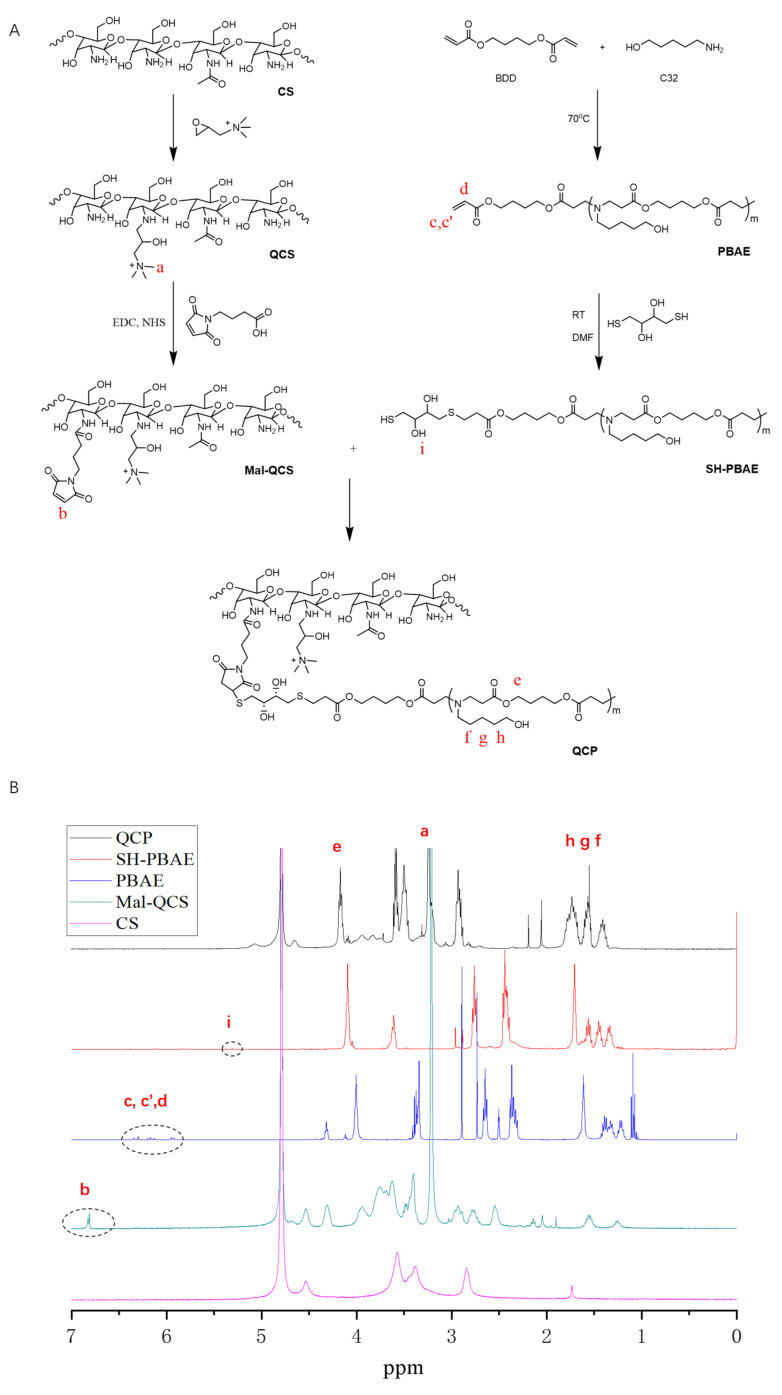
Synthesis and characterization of QCP. (**A**) Synthesis of QCP. (**B**) ^1^H-NMR of QCP and intermediate products.

**Figure 2 pharmaceutics-16-00154-f002:**
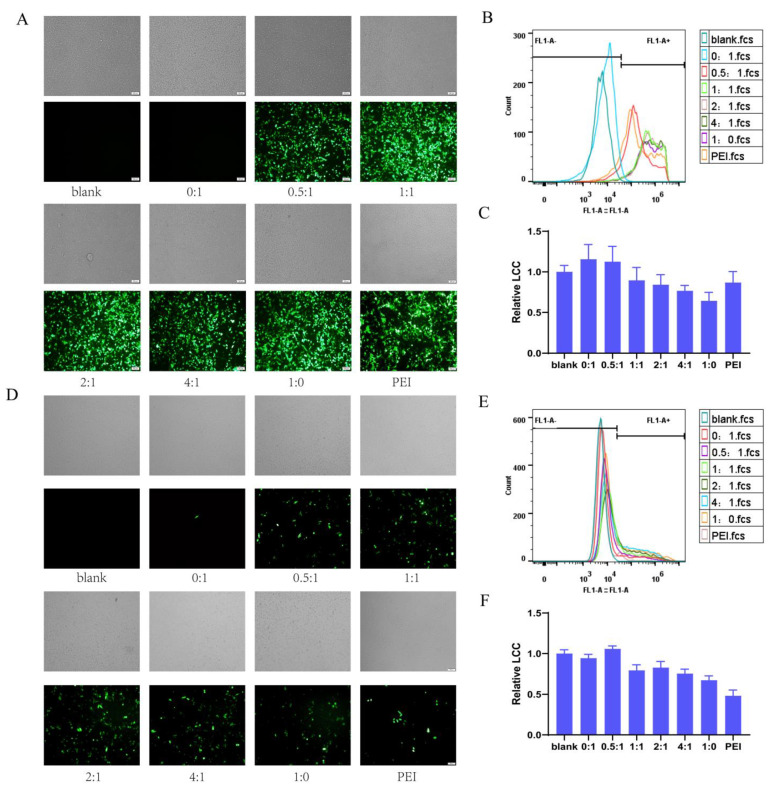
Transfection efficiency of PQ–GFP NPs on HEK293T and ME180 cells. Bright field (gray, for cell localization) and fluorescent images, cytometric analysis of transfection efficiency, and relative number of viable cells (LCC) of PQ–GFP NPs transfected HEK293T cells (**A**–**C**) and ME180 cells (**D**–**F**) at different mass ratios of PBAE and QCP for 36 h. Scale bar, 100 μm.

**Figure 3 pharmaceutics-16-00154-f003:**
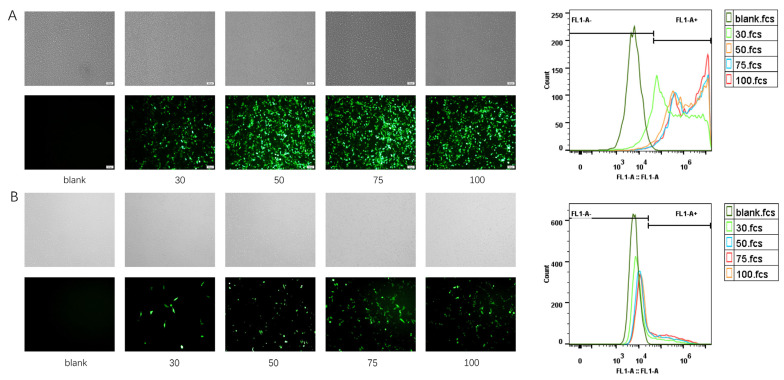
Bright field (gray, for cell localization), fluorescence images, and cytometric analysis of transfection efficiency of PQ–GFP NPs on HEK293T (**A**) and ME180 (**B**) cells at different mass ratios of PQ and GFP for 36 h with a mass ratio of PBAE and QCP of 2:1. Scale bar, 100 μm.

**Figure 4 pharmaceutics-16-00154-f004:**
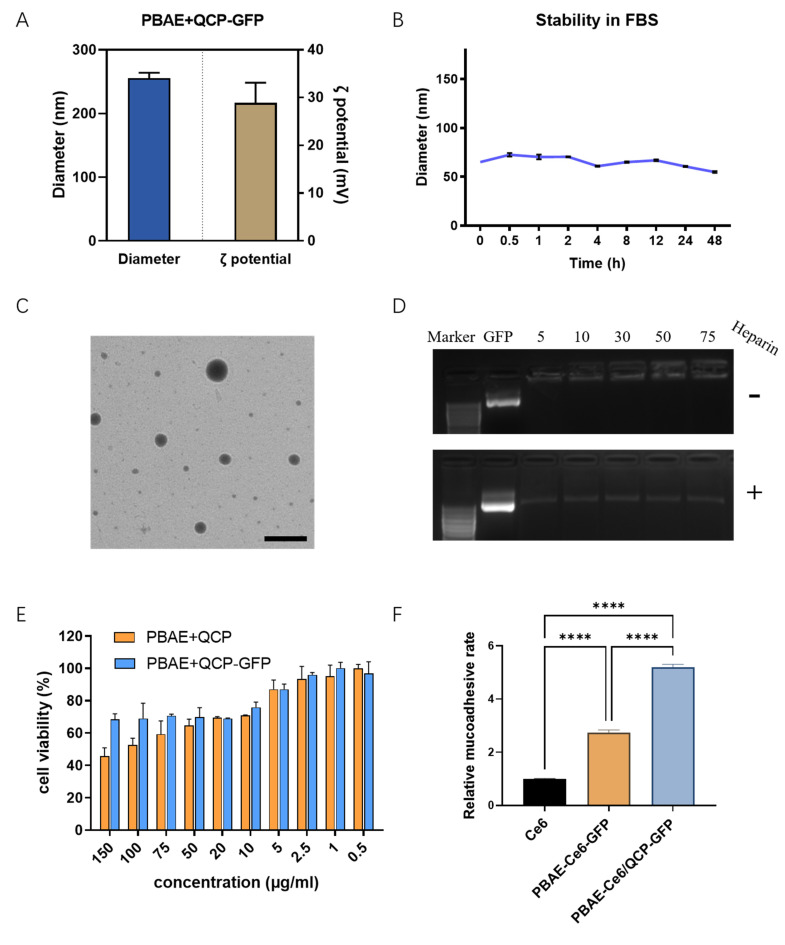
Characterization of PQ–GFP NPs. (**A**) Particle size and ζ-potential of PQ–GFP NPs. (**B**) Stability of PQ–GFP NPs in FBS at room temperature. (**C**) TEM image of PQ–GFP NPs. Scale bar, 300 nm. (**D**) Agarose gel electrophoresis pattern at different mass ratios of material and GFP with or without heparin. (**E**) Cytotoxicity of PQ–GFP NPs on ME180 cells. (**F**) Mucoadhesion rates of PBAE–Ce6/QCP–GFP NPs in the vagina of SD rats. Each point represents the mean ± SD (n = 3). One-way ANOVA was used for statistical analysis, **** *p* < 0.0001.

**Figure 5 pharmaceutics-16-00154-f005:**
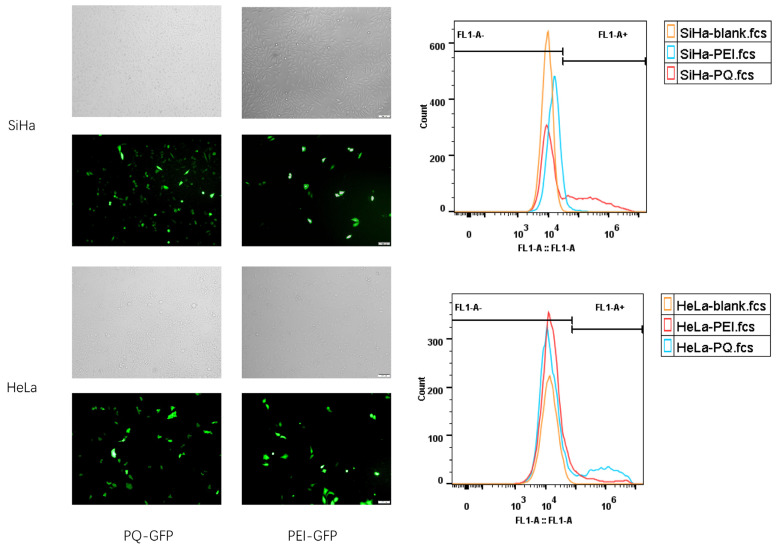
Transfection of PQ–GFP and PEI–GFP NPs in SiHa and HeLa cell lines. Scale bar, 100 μm.

**Figure 6 pharmaceutics-16-00154-f006:**
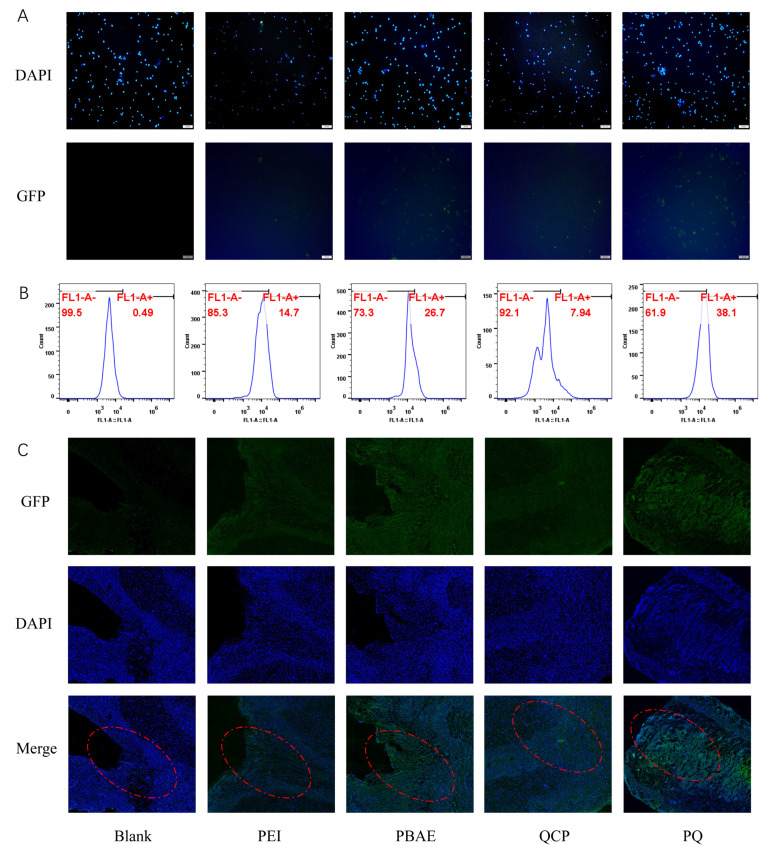
The in vivo transfection effect in SD rat vagina. (**A**) Fluorescence images and (**B**) flow cytometry analysis of the vaginal/cervix epithelial cells of SD rats treated with PEI, PBAE, QCP, and PQ–GFP NPs. Scale bars, 100 μm. (**C**) Frozen section images of SD rat vagina treated with different groups. Cervical/vaginal mucosa are marked by red circles. Scale bars, 200 μm.

**Figure 7 pharmaceutics-16-00154-f007:**
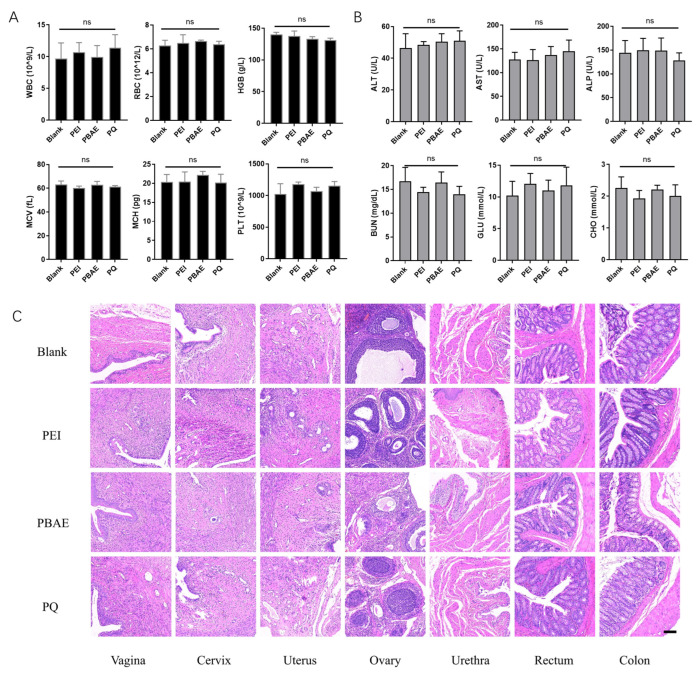
Toxicity analysis of PEI, PBAE, and PQ–GFP NPs in SD rats. (**A**) Routine blood and (**B**) blood biochemical tests of toxicity experiment for each group. Quantitative analysis of WBC, RBC, HGB, MCV, MCH, PLT, ALT, AST, ALP, BUN, GLU, and CHO. Each point represents the mean ± SD (n = 3). (**C**) Representative images of H&E staining of vagina and adjacent organs (cervix, uterus, ovary, urethra, rectum, and colon) in SD rats treated with PEI, PBAE, and PQ–GFP NPs. Scale bars, 100 μm. One-way ANOVA was used for statistical analysis; ns: no significant difference.

## Data Availability

Data are contained within the article and Appendix A.

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
