# Peer review of "Chitosan-Functionalized Poly(β-Amino Ester) Hybrid System for Gene Delivery in Vaginal Mucosal Epithelial Cells"

_pharmaceutics, 2024, doi:10.3390/pharmaceutics16010154_

Round 1

Reviewer 1 Report

Comments and Suggestions for Authors

The paper describes the transfection performance of a complex cationic co-polymer synthesized with chitosan to increase mucoadhesion. The GFP plasmid is used as a model. The work is similar to a paper previously published by the authors for the delivery of a therapeutic plasmid against HIV (ref 27). The paper is interesting and the conclusions are consistent with the results.

The experimental part is written in a summary way and should be improved (see subsequent comments). The captions of figures 2 and 3 need to be improved with more details (for example, it is not written what the gray non-fluorescent images represent and what meaning do they have)

-          Page 5 line 158. What do the authors mean by "immersed". If I understand correctly, the  vaginal tissue is immersed in different types of polymer but for how long time and at what concentration were the polymers?  At the end of the experiment, the fluorescence of what is read? Is the data obtained by difference with initial fluorescence? On which calibration curve?

-          Page 5 line 164. How many rats are used in the experiment. What does continuous application mean? Do authors mean that is a single application is left in situ for 36 hours?

-          Page 6 line 176 When were the analyzes carried out? At the beginning of the treatment, during or only at the end?

Author Response

Reviewer 1

The paper describes the transfection performance of a complex cationic co-polymer synthesized with chitosan to increase mucoadhesion. The GFP plasmid is used as a model. The work is similar to a paper previously published by the authors for the delivery of a therapeutic plasmid against HIV (ref 27). The paper is interesting and the conclusions are consistent with the results.

We sincerely thank Reviewer 1 for his/her time and thoughtful comments. Corresponding changes have been made in Red text in the revised submission, which are listed below point-by-point.

The experimental part is written in a summary way and should be improved (see subsequent comments). The captions of figures 2 and 3 need to be improved with more details (for example, it is not written what the gray non-fluorescent images represent and what meaning do they have)

Response: Thank you for the helpful suggestion. The gray images are bright field of the cells which represented the cell state and the localization of GFP (as there is no staining of the cell membrane). We revised captions of Figures 2 and 3.

Page 5 line 158. What do the authors mean by "immersed". If I understand correctly, the  vaginal tissue is immersed in different types of polymer but for how long time and at what concentration were the polymers?  At the end of the experiment, the fluorescence of what is read? Is the data obtained by difference with initial fluorescence? On which calibration curve?

Response: Thanks for the constructive comment. The vaginal tissue was indeed immersed in different types of polymer. The incubation time was 3 h at 37 ℃. The concentration of the polymers were 1 mg∙mL−1. Ce6 was excited at 633 nm, and emission was 660 nm. The data was relative quantitation obtained by the ratio of emission intensity of PBAE–Ce6/QCP–GFP NPs to that of free Ce6 after adhesion testing. We revised section 2.6 to clearly describe the test.

Page 5 line 164. How many rats are used in the experiment. What does continuous application mean? Do authors mean that is a single application is left in situ for 36 hours?

Response: Thank you for raising this important point. The experiment of in vivo transfection ability consisted of five groups, with 3 rats in each group. This information has been listed in Page 6, line 183. Continuous application means that the administration method for rats was once a day for 3 consecutive days, totaling three times. We revised the sentence in Page 6, line 185.

Page 6 line 176 When were the analyzes carried out? At the beginning of the treatment, during or only at the end?

Response: Thanks for the help comment. The toxicity analyses were carried out at the end of the experiment, on the 6th day. The detail was added to Page 6, line 193.

Reviewer 2 Report

Comments and Suggestions for Authors

The authors synthesized a quaternized chitosan (QCS)/poly(beta-amino ester) (PBAE) conjugate for application to gene delivery in vaginal mucosal epithelial cells. This material seems new, and the authors demonstrated efficient transfection activity in vivo as well as in vitro. Thus, this is worth to be published. However, there are several problems, as follow.

1)  The authors should clarify the significance and advantages of the hybrid system of QCS and PBAE in the Introduction.

2) Check the references in the Introduction. I think that some important papers about gene delivery using QCS and PBAE (eg. Mastorakos, 2015, PNAS) are lacking.

3) Comparison of the hybrid system to the complexes with QCS or PBAE is crucial to show the advantages. Although the in vivo results were clearly compared among them, the in vitro and physicochemical experiments for the complexes with QCS are lacking (or it is difficult to compare them). Add these results. Or, improve the description of figures and tables.

4) Although there are many papers to show the efficient transfection activity of the DNA complex with QCS, it was quite low in this study. Did the author optimize the ratio of DNA to QCS? The authors should add the discussion. 

5) Zeta potentials of each complex prepared different polymers at different ratios are important to understand the transfection efficiency.

6) Label of panel in imaging data should be shown in the upper side, not in the bottom side.

Comments on the Quality of English Language

Minor editing of English language seems required.

Author Response

Reviewer 2

The authors synthesized a quaternized chitosan (QCS)/poly(beta-amino ester) (PBAE) conjugate for application to gene delivery in vaginal mucosal epithelial cells. This material seems new, and the authors demonstrated efficient transfection activity in vivo as well as in vitro. Thus, this is worth to be published. However, there are several problems, as follow.

We sincerely thank Reviewer 2 for his/her time and thoughtful comments. Corresponding changes have been made in Red text in the revised submission, which are listed below point-by-point.

1)  The authors should clarify the significance and advantages of the hybrid system of QCS and PBAE in the Introduction.

Response: Thank you for the constructive advice. The significance and advantages of QCS and PBAE were affiliated in the Introduction (Page 2, line 61-74).

2) Check the references in the Introduction. I think that some important papers about gene delivery using QCS and PBAE (eg. Mastorakos, 2015, PNAS) are lacking.

Response: Thanks for raising the important point. The detailed explanations about QCS and PBAE, as well as related references, were provided in the Introduction (Page 2, line 61-74).

3) Comparison of the hybrid system to the complexes with QCS or PBAE is crucial to show the advantages. Although the in vivo results were clearly compared among them, the in vitro and physicochemical experiments for the complexes with QCS are lacking (or it is difficult to compare them). Add these results. Or, improve the description of figures and tables.

Response: Thanks for the reviewer’s reminding. We conducted the in vitro experiments about the transfection ability of CS and QCS, and the results showed that CS and QCS alone could hardly transfect GFP in the ME180 cell line (Figure S4). In addition, the comparison between the hybrid system and QCP or PBAE was fully demonstrated in the in vitro transfection experiment (Figure 2: PBAE:QCP = 0:1 for pure QCP, PBAE:QCP = 1:0 for pure PBAE). We added corresponding discussion and analysis in Section 4 (Page 14, line 351-380).

4) Although there are many papers to show the efficient transfection activity of the DNA complex with QCS, it was quite low in this study. Did the author optimize the ratio of DNA to QCS? The authors should add the discussion. 

Response: Thank you for the constructive comment. We investigated the transfection ability of QCS in ME180 cells with different QCS-GFP ratio (Figure S4). We have described the results and did some analysis with discussion, as shown in Discussion and Conclusions. More detailed explanations can be referred to Section 4 (Page 14, line 351-380).

5) Zeta potentials of each complex prepared different polymers at different ratios are important to understand the transfection efficiency.

Response: Thank you very much for the helpful comment. To further explore the correlation between ζ-potential and TE, we examined the potential of various PQ-GFP NPs, as detailed in Figure S5. The relevant descriptions can be referred to section 3.3 (Page 9, line 267-273) and section 4 (Page 14, line 365-370).

Figure S5. ζ-potential of PQ–GFP NPs at different mass ratios of PBAE and QCP (A) and mass ratios of PQ and GFP (B). Each point represents the mean ± SD (n = 3).

6) Label of panel in imaging data should be shown in the upper side, not in the bottom side.

Response: Thanks for the help comment. Label of panel in imaging data was displayed in the upper side, as shown in Figure 4D.

Figure 4D. Agarose gel electrophoresis pattern at different ratios of material and GFP with or without heparin.

Reviewer 3 Report

Comments and Suggestions for Authors

This manuscript (pharmaceutics-2780108) by Gao et al., introduced quanternized chitosan-modified poly(beta-amino ester) as a gene delivery systems for cervical mucosal epithelial cell transfection. Overall, I ask minor revision as below.

1. Please show the vector map of pGFP in the supplementary Figure.

2. Please also report the cationic polymer to GFP ratio as a N/P ratio in Figure 3

3. Is there any reason not to use PEG in the gene delivery systems?

4. Please indicate the statistical method in each figure legend.

Author Response

Reviewer 3

This manuscript (pharmaceutics-2780108) by Gao et al., introduced quanternized chitosan-modified poly(beta-amino ester) as a gene delivery systems for cervical mucosal epithelial cell transfection. Overall, I ask minor revision as below.

We sincerely thank Reviewer 3 for his/her time and thoughtful comments. Corresponding changes have been made in Red text in the revised submission, which are listed below point-by-point.

  1. Please show the vector map of pGFP in the supplementary Figure.

Response: Thanks for the reviewer’s reminding. The vector map of pGFP was presented in Figure S2. The relevant descriptions can be referred to Page 7, line 229.

Figure S2. The vector map of pGFP.

  1. Please also report the cationic polymer to GFP ratio as a N/P ratio in Figure 3

Response: Thanks for the constructive comment. The N/P ratios of different cationic polymers to GFP were all presented in Table S1, and the detail was added to Page 5, line 136-137.

  1. Is there any reason not to use PEG in the gene delivery systems?

Response: Thanks for raising this important point. During the implementation of gene therapy for cervical cancer, a deliver system that can overcome mucus barriers, prolong the drug retention time, and enhance therapeutic efficacy is desirable. Some studies did demonstrate that a high-density surface coating of hydrophilic and neutral charged polyethylene glycol (PEG) possessed a mucus-inert surface, making the complex NP resistant to mucous adhesion, thereby enabling rapid diffusion through airway mucus (Proc Natl Acad Sci U S A 2015, 112 (28), 8720-5). Chitosan and its derivatives can be used as mucoadhesive polymer for drug delivery via various mucosal surfaces. The mucoadhesiveness of chitosan is pH-dependent and stronger at the acidic pH (Eur J Pharm Biopharm 1999, 47 (3), 269). So we tried to introduce CS derivative as mucoadhesiveness enhancer to improve the local gene delivery ability of PBAE. We revised the introduction to state the reason we chose CS (Page 2, line 61-67). Thank you again for your constructive comment and we’ll synthesize PEG-PBAE copolymer to evaluate the vaginal gene delivery ability.

  1. Please indicate the statistical method in each figure legend.

Response: Thank you very much for the helpful comment. One-way ANOVA was used for statistical analysis in Figure 4. Moreover, we added statistical analysis in Figure 7A and B (Page 14, line 349).

Figure 7. (A) Routine blood and (B) blood biochemical tests of toxicity experiment for each group. Quantitative analysis of WBC, RBC, HGB, MCV, MCH, PLT, ALT, AST, ALP, BUN, GLU, and CHO. Each point represents the mean ± SD (n = 3). One-way ANOVA was for statistical analysis , ns: no significant difference.

Round 2

Reviewer 2 Report

Comments and Suggestions for Authors

The authors replied to my comments properly. However, some points remain to be improved, as follows.

1) Explanatory notes in FACS results are too small to read.

2) Bar graphs seem better in panels (C) and (F) of Figure 2.

3) It is better that the ratios of CS:QCP and polymer:GFP are clarified as weight ratios, mole ratios or volume ratios.

4) Discussion and conclusion parts should be separated.

Author Response

We sincerely thank Reviewer 2 for his/her time and thoughtful comments. Corresponding changes have been made in Red text in the revised submission, which are listed below point-by-point.

The authors replied to my comments properly. However, some points remain to be improved, as follows.

1) Explanatory notes in FACS results are too small to read.

Response: Thank you for the helpful suggestion. All the explanatory notes in FACS results have been revised, as shown in Figure 2B, 2E, 3A, 3B, 5 and 6B.

2) Bar graphs seem better in panels (C) and (F) of Figure 2.

Response: Thanks for your constructive comment. Bar graphs were presented in Figure 2C and 2F, and we revised the corresponding data in Table S1.

3) It is better that the ratios of CS:QCP and polymer:GFP are clarified as weight ratios, mole ratios or volume ratios.

Response: Thank you for raising this important point. The ratios of CS:QCP and polymer:GFP are mass ratios (weight ratios). We have clarified this in Introduction (Page 2, line 80) and other parts of the manuscript.

4) Discussion and conclusion parts should be separated.

Response: Thanks for your reminding. We have separated the discussion and conclusion (Page 14 ).
